# Pollution Characteristics of Microplastics in Soils in Southeastern Suburbs of Baoding City, China

**DOI:** 10.3390/ijerph17030845

**Published:** 2020-01-29

**Authors:** Chuan Du, Handong Liang, Zhanping Li, Jie Gong

**Affiliations:** 1State Key Laboratory of Coal Resources and Safe Mining, Beijing 100083, China; dc13298312753@163.com (C.D.);; 2College of Geoscience and Surveying Engineering, China University of Mining and Technology (Beijing), Beijing 100083, China; 3Key Laboratory of Organic Optoelectronics and Molecular Engineering of Ministry of Education, Department of Chemistry, Tsinghua University, Beijing 100084, China

**Keywords:** microplastics, soil, risk assessment, TOF-SIMS, pollution characteristics

## Abstract

Microplastics (MPs) are emerging pollutants that exist in different environmental media. Because of their wide range and large potential environmental hazards, they have attracted widespread attention in recent years. At present, the research on MP is mostly concentrated on the water ecosystems, and the impact on soil ecosystems is less studied. In this study, 12 typical soil samples from southeastern suburbs of Baoding city were investigated and characterized by time-of-flight secondary ion mass spectrometry (TOF-SIMS) combined with mass high resolution mode and positive and negative ion imaging mode. Four types of MPs, poly (propylene) (PP), poly (vinyl chloride) (PVC), poly (ethylene terephthalate) (PET), and poly (amide 6) (PA6), were quickly identified, of which PET and PA6 accounted for the largest proportion of both up to 30.2%; the particle size of the obtained MPs ranged from 0 to 35 μm, of which the proportion of <10 μm MPs was more than 26.3%, while that of 20–25 μm and 25–35 μm MPs was relatively small (17.83% and 9.3%, respectively). Risk assessment results of the MP in the soil showed that the risk level of MPs in the non-ferrous metal industrial parks and in concentrated with small workshops areas is relatively high, and attention should be paid to such areas. In addition, the study provides a reference method for the investigation and risk assessment of MPs in terrestrial soils, coastal beaches, and sediments.

## 1. Introduction

In recent years, microplastics (MPs), as a new type of environmental pollutant, have gradually attracted the attention of the public and of scholars [1,2,3,4]. MPs are usually defined as environmental plastic pollutants smaller than 5 mm, which exist widely in oceans, rivers, sediments, soils, and organisms [5,6,7,8]. Although MPs have attracted widespread attention in academic circles, most of studies have focused on marine MPs, and there are few studies on MPs in the terrestrial soil environment [9,10,11]. Until now, the research on soil environmental MPs has mainly focused on farmlands, such as research on farmlands in the suburbs of Shanghai and farmlands near Dianchi Lake [12,13]. Previous studies have examined the spatial distribution of MPs in riparian sediments, the soil microbial uptake, the sorption and bioaccumulation of persistent organic pollutants (POPs), and the transport of MPs in microbial communities and plants [14]. These effects may also be applicable to terrestrial soil systems, so more attention should be paid to the study of MPs in soils. A recent study conducted in the atmospheric environment confirmed that clothes drying in natural sunlight is a direct source of atmospheric suspension of MPs [15]. In freshwater ecosystems, Wang confirmed that anthropogenic factors affect the abundance of MPs [16]. To assess the environmental risks associated with MPs, different ecosystems should be studied.

At present, there are several ways to identify MPs in soil. Vianello identified MPs from the Venetian lagoon sediments of 1 mm or less by μFTIR, the most frequent size (93% of observed MPs) was in the range 30–500 μm, and up to 70% of particles was erroneously identified as MPs through FTIR analysis [2]. Sobhani implemented Raman to map MP towards their identification and visualisation, with a lateral resolution down to 1 μm, but the technique of Raman spectroscopy would be interrupted by the presence of color, additives and attached contaminants MPs [17]. SEM is also used for identification of MPs, and provides high-magnification and clearer structural images of MPs. Nevertheless, SEM detection takes a lot of time and is relatively expensive [18]. In addition, Pyr-GC/MS has been also used to gain structural information of polymer by analyzing their thermal degradation products. However, the materials analyzed by Pyr-GC/MS would be destructive [19]. Jungnickel used time-of-flight secondary ion mass spectrometry to identify PE in the ocean, which can quickly perform mass spectrometry and imaging on MPs. The sample preparation steps are simple, and the samples are less damaged [20]. Besides, Du used TOF-SIMS to characterize the MPs found in farmlands and analyze their distribution characteristics [21]. Up to now, there have been many method and index for environmental risk assessment of MP. Wilcox used species distribution models, numerical plastic modelling, and ingestion rates across a diverse range of marine species with different life history traits, to undertake a risk assessment for the Mediterranean Sea region [22]. Schuyler used vertebrate organisms to obtain the first empirical estimate of the relationship between plastic ingestion and death [23]. Based on the chemical composition of plastic polymers, Lithner incorporated additives, chemical hazards of monomers, polymers, and polymers, and developed a hazard grading model for plastic polymers to evaluate the impact of hazards on human health and biology [24]. The southeastern suburbs of Baoding city are located in the central functional area of the Beijing-Tianjin-Hebei coordinated development. The study area is an important manufacturing base in northern China, and it is a typical urban-rural integrated area represented by small processing enterprises. The northern part of the study area is 162 km away from Beijing, 130 km away from Tianjin, and 187 km away from Shijiazhuang in the north. The soil types are mainly fluvo-aquic soil and moist soil, and the soil texture is mainly sandy loam and loam. Most of the county is rural, which accounts for more than 80%, and the main land use type is farmland. The study area is known as the North China Shoe Capital and has more than 1000 shoe processing and manufacturing workshops. These small workshops not only consume a lot of energy, but also produce a large number of industrial waste and toxic gases. At the same time, the area also includes plastic processing and down industries. The processing of plastic products consumes a lot of resin and produces a large amount of industrial waste, most of which is discharged into nearby farmland and rivers. In this study, time-of-flight secondary ion mass spectrometry (TOF-SIMS) was used to characterize the MPs in urban-rural integration area soils of northern China, and its distribution characteristics were analyzed. Meanwhile, the risk scoring method of plastic polymers created by Lithner was adopted to evaluate the environmental risk of the MPs. We hoped that this study can assist policy formulation of MP pollution in terrestrial soils by providing preliminary data on MPs assessment.

## 2. Materials and Methods

### 2.1. Sample Collection

The study area was located near Baoding City in Hebei Province, northern China (38°70′–38°96′ N, 115°67′–116°13′ E). According to the different urban functional areas, 12 typical areas were selected (Table 1). Soil samples at a depth of about 10 cm from the surface were collected using a stainless steel sampling shovel. At least five soil samples (within 100 m^2^) were collected at each sampling point; the samples were mixed and homogenized into a composite sample, wrapped with aluminum foil, put into a sampling bag, brought back to the laboratory, placed in a clean and light-proof place, dried, and preserved at low temperature.

### 2.2. Sample Pretreatment

At present, there are few detection methods for MPs with a particle size of 1–20 μm, and the maximum scanning area of TOF-SIMS is 500 μm^2^. The large particle size is not conducive to imaging characterization. It is considered that TOF-SIMS is suitable for detecting MPs smaller than 35 μm. After air drying, the soil samples were screened using a 35 μm sieve, and the preliminary separation samples (about 100 g) were obtained. About 1 g (dry weight) was weighed for MPs extraction. In total, 4 mL of 30% H_2_O_2_ solution was added, then added to 10 mL ultrapure water, and the samples were sealed. The sample was then incubated at 60 °C constant temperature water bath for 6 h to remove natural organic matter, and the sample was left for 1 h to ensure aggregates dispersed in the sample. After standing for 24 h, a small amount of supernatant liquid was absorbed on the silicon chip by a liquid transfer gun, and then, the sample was put into a fume hood to dry naturally for 24 h. Before the experiment, the samples were stored in Petri dishes and kept away from light. According to this method, 12 soil samples were collected to be processed in turn.

### 2.3. TOF-SIMS

TOF-SIMS analysis was carried out on a TOF-SIMS 5 (ION-TOF GmbH, Münster, Germany), equipped with a Bi liquid metal ion gun (LMIG). TOF-SIMS spectra and images were acquired using a 30 keV Bi_3_^++^ LMIG. The Bi_3_^++^ current was 0.8 pA (<1 ns pulse width, bunched beam). Charge compensation was accomplished using a flood gun with low-energy electrons. TOF-SIMS spectra and images were acquired using a 30 keV Bi_3_^++^ LMIG. The Bi_3_^++^ current was 0.8 pA (<1 ns pulse width, bunched beam). Image of a single MP was acquired using Bi_3_^++^ LMIG rastering over a 500 × 500 μm^2^ area with 128 × 128 pixels in normal 2D image mode. The total Bi_3_^++^ accumulated ion dose was between 1011 and 1012 ions/cm^2^. The Bi_3_^++^ LMIG was operated at a 200 μs cycle time (mass range: 0–2933 u). Positive spectra were mass-calibrated using CH_3_^+^, C_2_H_3_^+^, C_2_H_5_^+^, C_3_H_7_^+^, C_4_H_9_^+^, and C_5_H_11_^+^. The mass resolutions (measured at C_2_H_5_^+^, *m*/*z* = 29) were typically >6000.

### 2.4. Standard Sample Analysis

TOF-SIMS analyzed six common MPs standard samples, including poly (ethylene) (PE), poly (propylene) (PP), poly (styrene) (PS), poly (vinyl chloride) (PVC), poly (amide 6) (PA6) and poly (ethylene terephthalate) (PET) [16,18,20,25,26,27,28,29]. PE pellets, PP pellets, PS pellets, PVC pellets, PA6 and PET pellets (about 30 μm) were bought from Sigma-Aldrich (Milwaukee, WI, USA). For ion detection, standard samples of PE, PP, PS, PVC, PET, and PA6 were first tested, and then, the mass spectra of each standard sample were compared. However, only the typical fragments of PA6, PET, PVC, and PP could be distinguished from each other (Table 2). Therefore, these four MPs were considered suitable for identification in fresh soil samples.

### 2.5. Risk Assessment

In this study, the chemical composition and relative abundance of MPs were considered to assess the potential risk of MPs in surface soil of the study area. Following Lithner et al. the chemical toxicities of MP polymers were used as an important indicator for assessing their ecological hazards. Therefore, the hazard score of Lithner’s plastic polymers and the type of polymers used to evaluate the risk of MPs were used in this study (Table 3) [24]. The formula is as follows:(1)H=∑Pn×Sn
where *H* is the calculated polymer risk index of MPs, *Pn* is the percentage of different types of MPs collected at each sampling point, and *Sn* is Lithner’s score for a polymer compound consisting of MP particles.

This study provides a classification standard of risk level of MPs (Table 4), which can initially be used to understand the risk level of MP pollution in the study area and also to provide support for risk management of MP pollution in soil [30,31].

### 2.6. Data Analysis

Using the respective peak lists as described above the raw data sets were further processed using ION-TOF Surface Lab 6.6. Particle counting and size measurement was executed with Nano Measurer 1.2.

## 3. Results and Discussion

### 3.1. TOF-SIMS Analysis

Taking XA1 sample as an example, the typical fragments of four MPs were analyzed by ion mass spectrometry, and the representative data of PP, PVC, PET, and PA6 were collected with imaging information. Accordingly, the characteristic ions of MPs in were obtained (Figure 1). In addition, the characteristic ions of clay minerals such as Na^+^, Al^+^, Si^+^ and Ca^+^ were also detected on the surface of MPs (Figure 2o–r).

The color shading presented by the ion imaging mass spectrometry indicates a change in the magnitude of the ion count values at different sites and is a reflection of the change in the concentration distribution of specific ion composition in the 500 μm × 500 μm microzone (Figure 2). The characteristic ions corresponding to PP, PVC, PET, and PA6 were all present as single particles with certain morphologies in the target region, and the relative signal intensity was high. The trend of changes differed among the characteristic ions of all MPs, thus denoting that the characteristic ions of these four MPs can be distinguished from each other. To a large extent, this information also shows that most of the MPs in farmland soil in this area are single polymers. C_4_H_9_^+^, C_5_H_11_^+^, C_6_H_9_^+^, C_6_H_11_^+^, and C_7_H_11_^+^ are all characteristic ions of PP and have similar light and dark changes in the lower left quarter (Figure 2a–e), but C_5_H_11_^+^ can distinguish PP from other polymers and substrates more effectively than other characteristic ions. It is clear that *m*/*z* = 71.083 (Figure 2b) for (C_5_H_11_^+^) as the main characteristic ion of PP. Similarly, the main characteristic ion of PA6, PET, and PVC should be *m*/*z* = 30.034 (CH_4_N^+^) (Figure 2f), *m*/*z* = 149.027 (C_8_H_5_O_3_^+^) (Figure 2k), and *m*/*z* = 83.982 (C_4_HCl^+^) (Figure 2l), respectively.

Figure 3a displays the overlay ion images using the most characteristic ions of the four polymers, C_5_H_11_^+^ (PP) in red, CH_4_N^+^ (PA6) in green, C_8_H_5_O_3_^+^ (PET) in blue, and C_4_HCl^+^ (PVC) in yellow. Thus, the distributions of the four types of MPs in the target area and the morphological information of the individual MPs could be clearly observed. These results show that TOF-SIMS analysis can distinguish different types of MPs in soil. Similar results can also be obtained for the main characteristic ions for XA2–XA12 sites by ion superposition plots (Figure 3b–l).

### 3.2. Type, Particle Size, Abundance and Distribution of MPs

The particle size and quantity information of the four types of MPs in soil can be obtained by using Nano Measurer 1.2 software to deal with the superposition map. In this study, a stainless-steel screen with a pore diameter of 35 μm was used for the preliminary separation, so the particle size of the MPs obtained ranged from 0 to 35 μm. The overall particle size distribution is shown in Figure 4b. More than 49% of the total MP particles were smaller than 15 μm; 26.4% and 23.3% were in the range of 0–10 μm and 10–15 μm, respectively. Very few MPs larger than 25 μm were observed, accounting for only 9.3% of the total MP concentrations. The proportion of MPs with different particle size ranges in the study area is 0–10 μm > 10–15 μm ≈ 15–20 μm > 20–25 μm > 25–35 μm. This is consistent with the results of most studies which state that the relative abundance of MPs decreases with increasing particle size. Because of the ubiquity of MPs smaller than 25 µm and their potential environmental significance over large size MPs, many studies have focused on MPs with particle sizes below 25 μm, such as in the mangrove wetlands of Singapore, where the particle size range of MPs is mainly below 20 μm [32]. There were significant differences in particle size distribution of MPs among the sampling sites. The MPs at XA1, XA5, XA7, XA8, XA9, and XA10 were mainly lower than 15 μm probably because the land use types of these sampling areas are mostly farmlands. In the past few decades, sewage sludge is usually recycled to use as fertilizer in farmland soil. The plastic concentrations in sludge were assayed in the range of 1500 to 24,000 items kg^−1^ [33,34,35,36]. Sewage sludge treated by a sewage treatment plant usually has a high abundance of MPs and a small particle size (μm level). At the same time, there are a large number of plastic microbeads in pesticides and fertilizers, namely, the original MPs. Through decades of agricultural activities, a large number of MPs with smaller particle sizes have accumulated in the region. Plastic products in soil are easier to decompose into MPs with smaller particle size under the effects of agricultural activities, ultraviolet radiation, and microbial decomposition [37,38,39]. However, the MPs at XA2, XA3, XA4, and XA11 were mainly larger than 20 μm. This may be because the MPs in these areas are less affected by external disturbances, and the MPs discharged in the environment are accumulated continuously, making further degradation more difficult.

The abundance of different types of MPs is different (Figure 4b). In general, the main types of MPs in this area are PA6 and PET, which account for more than 30% of the detected MPs, followed by PP, which accounts for 23.3%, and PVC, which accounts for 16.3%. Thus, the main MPs in the soil in this area are PA6 and PET. There are many small workshops in this area, most of which develop the down industry of shoes and clothing, and PA6, as one of the main raw materials of shoes and clothing production, produces a large amount of plastic waste in the processing and production processes. The non-ferrous metal industries are a local pillar industry, and related supporting industries such as pipe processing, automotive parts, hardware products, and other enterprises. Related industries in the non-ferrous metals industry, such as auto parts factories and electronic device factories, require a large amount of PET powder in the production process, and some PET waste will be directly discharged into the nearby soil, even with the rainwater runoff and soil sedimentation MPs will be discharged to Farther areas [40]. Therefore, PA6 and PET are the most common in the soil of this area.

Comprehensive analysis of the type and particle size of MPs shows that the particle size composition of different types of MPs is quite different (Figure 4c). PET and PVC have a higher abundance in particle sizes less than 15 μm, PA6 has a higher proportion in the 15–20 μm MPs, and PP has a uniform distribution in each particle size distribution range. Due to the poor toughness of PET, it exhibits certain brittleness and poor alkali resistance, so its properties in the soil are unstable and it is more likely to be broken and decomposed. Smaller MPs (μm grade) are easy to adsorb POPs (such as organochlorine pesticides, polycyclic aromatic hydrocarbons, and their derivatives) in soil and are more likely to enter biological tissues and even cells [6,41,42]. Therefore, in the future, more attention should be paid to these MPs with higher abundance and finer particles.

### 3.3. Chemical Risk Assessment of MP

According to the risk index (PI) of chemical characteristics of MP, the study area is facing serious MP pollution. The classification of MP pollution risk index is shown in Table 4. MP risks that cannot be ignored have been found in different urban functional areas of the study area (Figure 5). The high chemical risk index in some areas is due to the presence of MPs (such as PVC) with high hazard scores [30,31]. It can be seen from the figure that the risk level of MP pollution in the northern part of the study area is low, while the risk level of MP in the southern part of the study is relatively high, which is related to the distribution of urban functional areas in the study area. The northern functional area is the central urban area and commercial area and is away from the garbage treatment plant. Plastic waste that is closer can be processed centrally, so this area is less polluted by MPs. The southern and eastern regions are non-ferrous metal industrial parks and small workshops. During production and processing, a large amount of plastic waste is generated and most of it is discharged directly into the environment without treatment. Therefore, the risk level from MPs in the southern and eastern regions is relatively high.

Different from other refractory organic pollutants, the distribution of MPs in soil is uneven. Particle size and chemical composition of MPs are important factors affecting their distribution. More than 80% of the MPs studies here have similar shape characteristics, so they can be used to compare the risk of MPs in different regions. Some studies have also used similar risk assessment methods in the MP risk assessment of Shanghai river sediments and Yangtze River Estuary waters, which showed that high chemical toxicity can result in a high environmental risk of MPs [30].

In view of the chemical properties of MPs, it is believed that polymers are inert and often do not pose environmental risks to the soil environment [43]. However, plastic is not a single polymer, and unreacted monomers and other additives exist in plastics that are released into the environment as plastics decompose, posing a threat to ecosystems and human health. Polymers in MPs may also pose a threat to the environment. PVC is the most hazardous polymer and one of the most widely used plastic raw materials, with a global production of about 38.5 million tons [24]. Once PVC enters the soil, it releases carcinogenic monomers and other additives such as plasticizers, stabilizers, pigments, thus seriously affecting soil microorganisms and plants. At the same time, MPs also can adsorb POPs and transfer them to aquatic and terrestrial organisms, resulting in the absorption and storage of POPs in organisms to produce a complex ecological effect [44,45].

To date, MP contamination of terrestrial soils has not been part of environmental risk assessment. With the increasingly serious problem of MP pollution, the analysis of its distribution characteristics will help to better assess the environmental risk of MP, while gradually improved risk assessment methods and indicators will help to fully understand its impact on the ecological environment and on human health [30,46].

## 4. Conclusions

In this study, four kinds of MPs (PP, PVC, PET, PA6) in the surface soil of Baoding Suburbs were characterized by TOF-SIMS, and their pollution risks were preliminarily assessed. Twelve surface soil samples from various functional areas contained different degrees of MPs. The particle size of MPs in this area is mainly between 0 and 10 μm, and PA6 and PET account for the highest proportion. PVC is an important source of high pollution risk for MPs. According to the chemical risk assessment index for MP, non-ferrous metal industrial parks and small workshops were identified as the most polluted areas, reaching the highest risk level of V. Meanwhile, this study shows that the risk level of MP pollution is closely related to human production activities, and the region should be closely monitored. At the same time, MP evaluation indicators still need to be increased, which is the basis of future risk assessment. It is suggested that the academic circles should carry out the work on environmental risks of MPs as soon as possible to help control MPs pollution.

## Figures and Tables

**Figure 1 ijerph-17-00845-f001:**
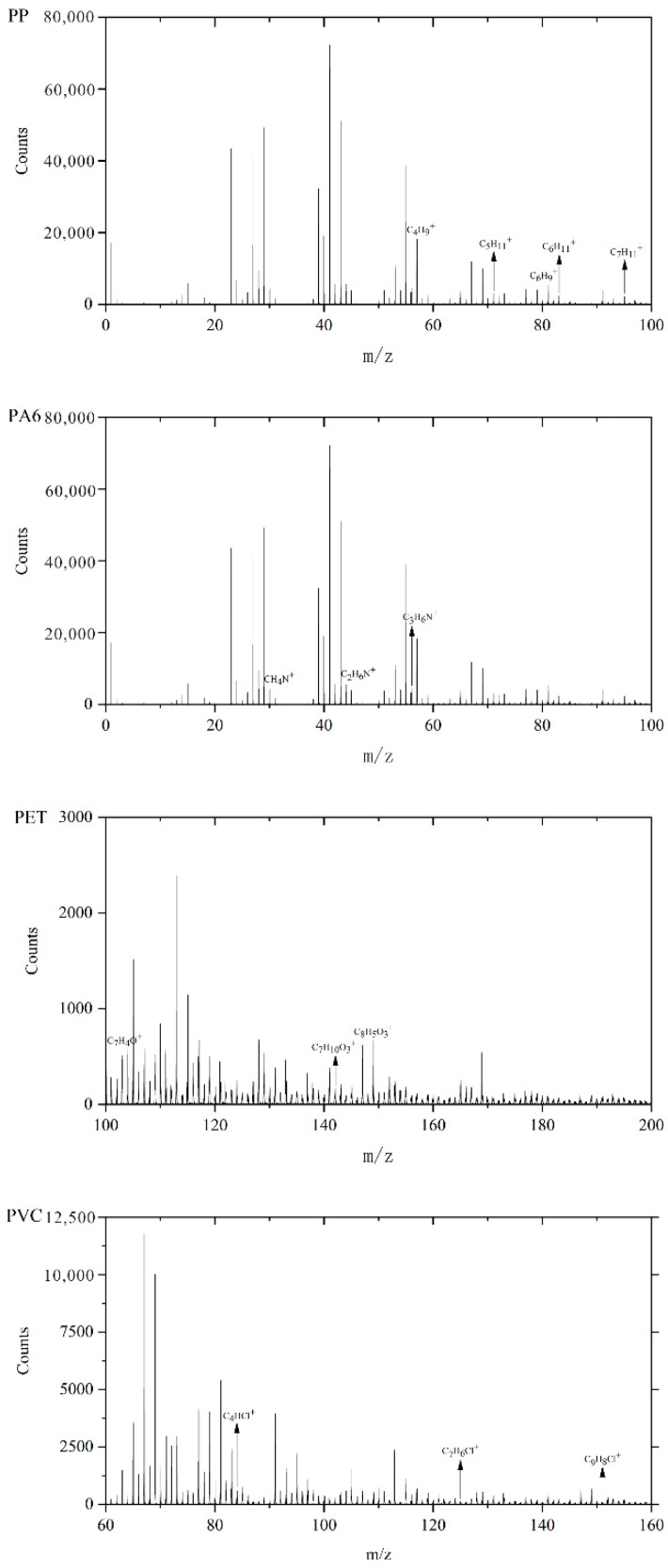
Positive ion spectra obtained XA1.

**Figure 2 ijerph-17-00845-f002:**
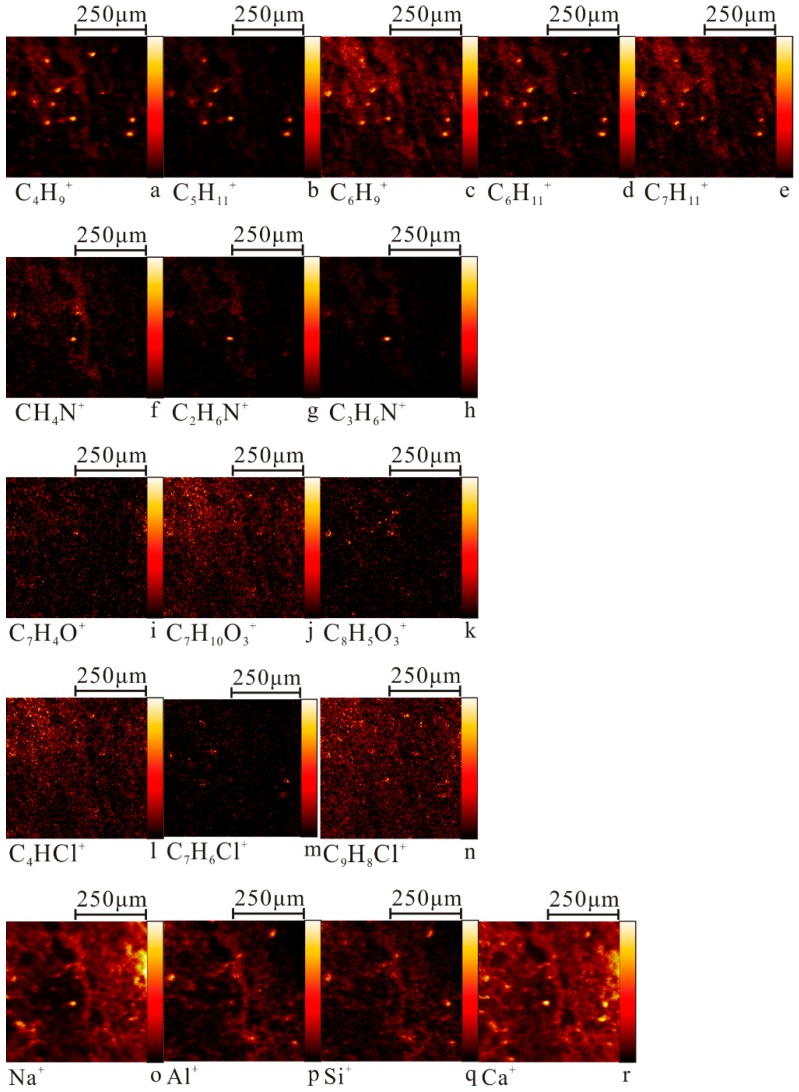
Time-of-flight secondary ion mass spectrometry (ToF-SIMS) maps of some characteristic ions of MPs in XA1 (**a**–**e**) PP; (**f**–**h**) PA6; (**i**–**k**) PET; (**l**–**n**) PVC.

**Figure 3 ijerph-17-00845-f003:**
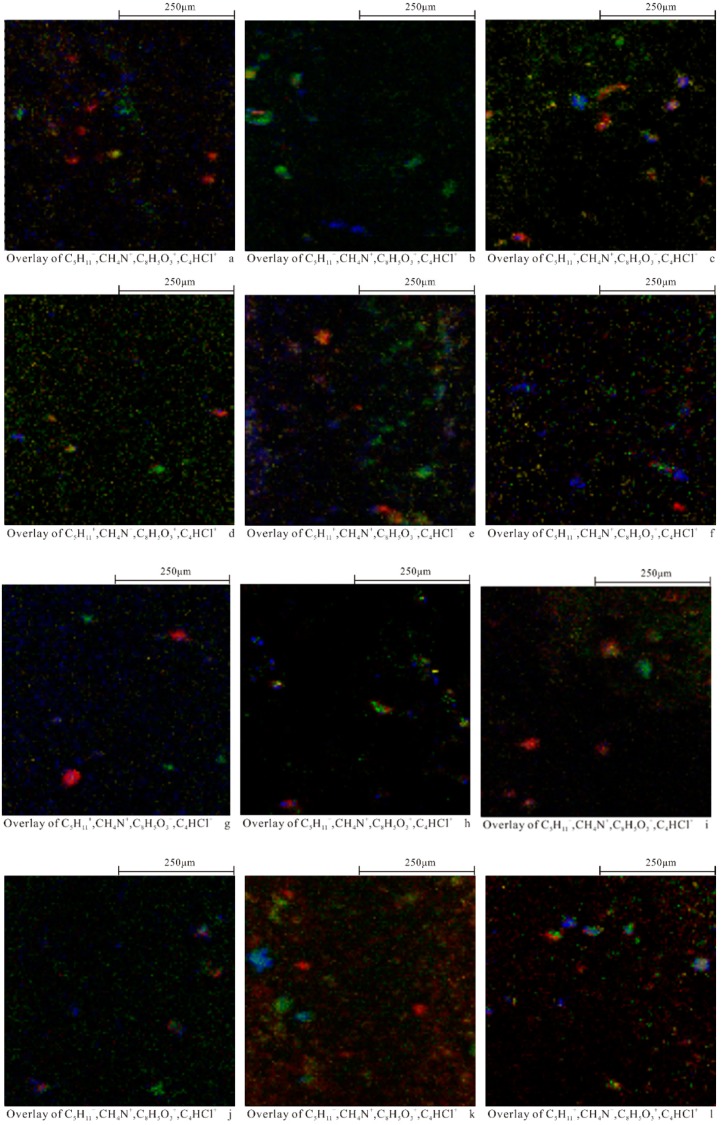
Ion-imaging superimposed images of four MPs (**a**) XA1; (**b**) XA2; (**c**) XA3; (**d**) XA4; (**e**) XA5; (**f**) XA6; (**g**) XA7; (**h**) XA8; (**i**) XA9; (**j**) XA10; (**k**) XA11; (**l**) XA12; 500 μm × 500 μm, 128 × 128 pixel.

**Figure 4 ijerph-17-00845-f004:**
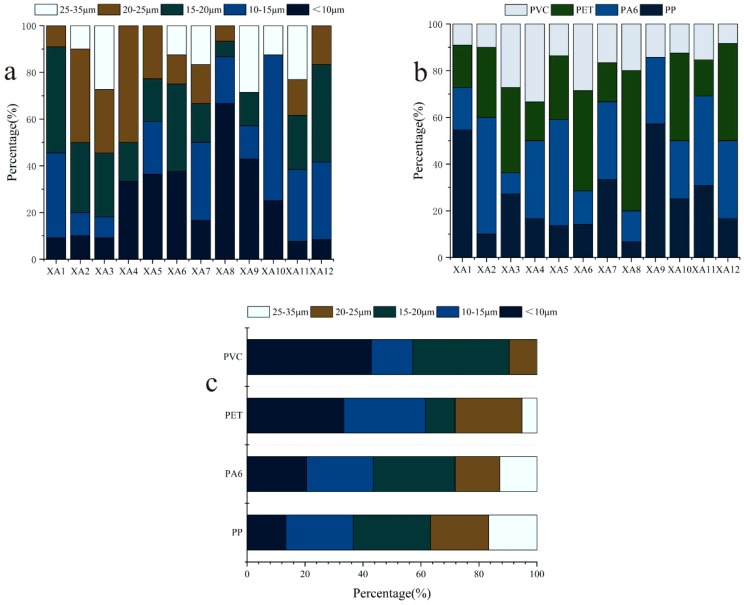
Particle size, type and abundance of MPs in soil. (**a**) Particle size composition and relative content of MPs in soil; (**b**) Composition and relative content of MPs types in soils; (**c**) Particle size composition of different types of MPs in the soil.

**Figure 5 ijerph-17-00845-f005:**
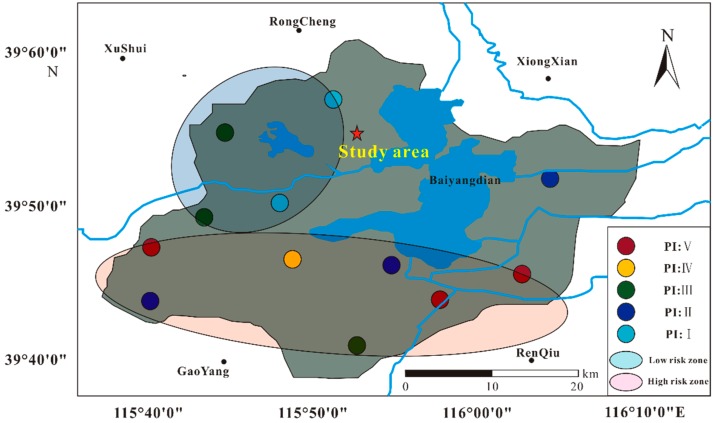
Risk degree of MPs in the study area. Different colors represent different polymer risk indexes.

**Table 1 ijerph-17-00845-t001:** Location of sampling points and soil utilization type.

Site Number	Latitude and Longitude	Elevation	Soil Utilization Type
XA1	115.829121° E, 38.793376° N	11 m	Industrial Park
XA2	115.667520° E, 38.737593° N	4 m	Factory
XA3	115.967873° E, 38.737334° N	14 m	Forest Land
XA4	116.129161° E, 38.922821° N	14 m	Residential Area
XA5	115.967832° E, 38.774490° N	11 m	Factory
XA6	115.667495° E, 38.792877° N	22 m	Farmland
XA7	115.921591° E, 38.700302° N	9 m	Factory
XA8	115.852379° E, 38.792931° N	7 m	Residential Area
XA9	115.713766° E, 38.848288° N	7 m	Farmland
XA10	116.105875° E, 38.903642° N	4 m	Farmland
XA11	115.760000° E, 38.922257° N	11 m	Factory
XA12	115.898340° E, 38.957338° N	5 m	Farmland

**Table 2 ijerph-17-00845-t002:** Typical fragment ions of four microplastic MP standard samples.

Polymer Type	Characteristic Peak Mass (*m*/*z*)	Typical Fragment Ion Composition
PP	57.074, 71.091, 81.103, 83.134, 95.137	C_4_H_9_^+^, C_5_H_11_^+^, C_6_H_9_^+^, C_6_H_11_^+^, C_7_H_11_^+^
PA6	30.036, 44.051, 56.068	CH_4_N^+^, C_2_H_6_N^+^, C_3_H_6_N^+^
PET	104.014, 142.075, 149.006	C_7_H_4_O^+^, C_7_H_10_O_3_^+^, C_8_H_5_O_3_^+^
PVC	83.978, 125.050, 151.79	C_4_HCl^+^, C_7_H_6_Cl^+^, C_9_H_8_Cl^+^

**Table 3 ijerph-17-00845-t003:** Polymer information for MPs detected in this study.

Polymer Type	Monomer	Density (g/cm^3^)	Main Application	Fraction
PP	Propylene	0.85–0.94	Agricultural film, bottle, food packaging etc.	1
PA6	Caprolactam	1.13	Bearings, automotive applications, etc.	50
PET	Terephthalic acid and ethylene glycol	1.33–1.38	Food and drug packaging film, packaging bottle, automobile application, etc.	-
PVC	Vinyl chloride	1.41	Pipe, cable insulation, etc.	10,551

PET lacks ecological toxicity data, therefore its hazard score cannot be determined. The value for the score of each polymer is taken from Lithner [24].

**Table 4 ijerph-17-00845-t004:** Risk rating criteria for MP contamination.

Value of Polymer Index	<1000	1000–1500	1500–2000	2000–2500	>2500
Risk category (PI)	Ⅰ	Ⅱ	Ⅲ	Ⅳ	Ⅴ

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
