# Peer review of "Pollution Characteristics of Microplastics in Soils in Southeastern Suburbs of Baoding City, China"

_ijerph, 2020, doi:10.3390/ijerph17030845_

Round 1
Reviewer 1 Report
In this manuscript, the authors investigated microplastic pollution in soil samples. Especially, the authors used "TOF-SIMS" method. The spectrapic method has been widely used in identification of microplastics. It is necessary to validate the new method before we use it. Though the authors cited one reference in the text, there was no the cited refeference in the reference list. I suggest that the authors provide adequate evidecs to prove the reliablity of this method in identifing polymers.
1. In Table 3, did the authors cite this table from other's reference? The cited paper should be indicated.
2. In Figures 2 and 3, the authors need to give some more explanation or other form of graphs to present results. It is hard for the readers in environmental science field to understand some original photographs. These figures look like raw data.
3. Generally, it is still a great challenge to identify microplastics less than 10 um. Since the authors indicate that they have identified 0-15 um particles, they need to provide reliable evidences.
4. In Figure 4, there are so much information in the same figure. Especially, "size" was presented in three different ways, which makes the readers confused and boring. It is better to present the most important results.
5. In Figure 5, it is hard to classify the risk in such a big area since there are only limited sampling sites.
Author Response
Response to Reviewer 1 Comments
Thanks for your comments on our paper. We have revised our paper according to your comments. We have added the reference cited in the text but not in the reference list to the reference list. For this new method of TOF-SIMS analysis of microplastics in soil, We have written a methodological article (Du, C., Liang, HD, Li ZP, Gong, Jie., 2020. ToF-SIMS Characterization of microplastics in soils. Surf. Interface Anal. Doi: 10.1002 / sia.6742.), which has just been received and is now added to the reference list for this manuscript. We believe the two articles cited can prove the reliability of the method in identifying microplastics. We hope that you find the revised manuscript acceptable for publication。
Point 1: In Table 3, did the authors cite this table from other's reference? The cited paper should be indicated.
Response 1: Yes, We have cited this table from other references. I have indicated the cited papers in the table and marked the manuscript in red.
Point 2: In Figures 2 and 3, the authors need to give some more explanation or other form of graphs to present results. It is hard for the readers in environmental science field to understand some original photographs. These figures look like raw data.
Response 2: For Figures 2 and 3, these are some images after the original data has been collated and modified. In order to facilitate the understanding of readers in the field of environmental science, We have given some more explanations in the manuscript.
Point 3: Generally, it is still a great challenge to identify microplastics less than 10 um. Since the authors indicate that they have identified 0-15 um particles, they need to provide reliable evidences.
Response 3: For currently widely used technologies, it is still a great challenge to identify microplastics less than 10 um. But using TOF-SIMS to identify microplastics less than 10 um is very fast and reliable. There are two papers that can prove the reliability of this new method. (Jungnickel, H., Pund, R., Tentschert, J., 2016. Time-of-flight secondary ion mass spectrometry (ToF-SIMS) -based analysis and imaging of polyethylene microplastics formation during sea surf simulation. Sci. Total. Environ. 563, 261-266.
Du, C., Liang, H.D., Li Z.P., Gong, Jie., 2020. ToF-SIMS characterization of microplastics in soils. Surf. Interface Anal. Doi: 10.1002 / sia.6742.)
Point 4: In Figure 4, there are so much information in the same figure. Especially, "size" was presented in three different ways, which makes the readers confused and boring. It is better to present the most important results.
Response 4: We agree with this comment of the reviewer. We have modified Figure 4 according to your suggestions.
Point 5: In Figure 5, it is hard to classify the risk in such a big area since there are only limited sampling sites.
Response 5: We agree with this comment of the reviewer. Due to limited time and funding, we collected limited samples and were only able to conduct a preliminary assessment of microplastics risks in the region. We hope that the statistical and risk assessment methods used in this study can provide references for other researchers.

Reviewer 2 Report
The aim of the study was to examine 12 typical soil samples from the south-eastern suburbs of the city of Baoding. Four types of microplastics: polypropylene (PP), poly(vinyl chloride) (PVC), poly(ethylene terephthalate) (PET), and polyamide 6 (PA6) have been identified. In addition, the study provides a reference method for the investigation and risk assessment of microplastics in terrestrial soils, coastal beaches, and sediments. The paper is an interesting study that can contribute to the scope of the International Journal of Environmental Research and Public Health. The manuscript contains only typographical mistakes which should be corrected throughout:
Missing space between words or extra space: page 1, line 3; page 2 lines 11, 15, 25; page 3 lines 3, 5, 8; page 5, line 17; Table 2; page 7, line 14; page 8, line 10, page 9, lines 7, 14; page 10, lines 2, 4, 5; page 15, lines 4, 5, 6; page 16, line 7, 22. “micro-plastics” – should be as before “microplastic”. Please explain the abbreviations the first time you use them: page 2, lines 19, 25; page 3, line 22. Table 1: Please remove the dots after “Factory.” and comma at page 5, line 9. Page 5, line 9 “waterbath” – these are 2 words. Please correct. Page 1, line 22; page 6, lines 12 and 13, Tables 2 and 3: “polyvinyl chloride” and “polyethylene terephthalate”. Please use IUPAC rules concerning polymer names. If the name of the monomer consists of more than one word, then it is parenthesized. Should be “poly(vinyl chloride)”and “poly(ethylene terephthalate)”. Please correct. Table 2 is illegible (specifically the 3rd column). Also Figure 1 and inscriptions under the Figure 3. Please correct. Page 14 line 5: “kg−1” – 1 should be in superscript. Please correct. For Figures and Tables, please use one style appropriate for this Journal (see page 16, lines 10, 11 and 23).Author Response
Response to Reviewer Comments
Thanks for your comments on our paper. We have revised our paper according to your comments。
Point 1: Missing space between words or extra space: page 1, line 3; page 2 lines 11, 15, 25; page 3 lines 3, 5, 8; page 5, line 17; Table 2; page 7, line 14; page 8, line 10, page 9, lines 7, 14; page 10, lines 2, 4, 5; page 15, lines 4, 5, 6; page 16, line 7, 22. “micro-plastics” – should be as before “microplastic”. Please explain the abbreviations the first time you use them: page 2, lines 19, 25; page 3, line 22. Table 1: Please remove the dots after “Factory.” and comma at page 5, line 9. Page 5, line 9 “waterbath” – these are 2 words. Please correct. Page 1, line 22; page 6, lines 12 and 13, Tables 2 and 3: “polyvinyl chloride” and “polyethylene terephthalate”. Please use IUPAC rules concerning polymer names. If the name of the monomer consists of more than one word, then it is parenthesized. Should be “poly(vinyl chloride)”and “poly(ethylene terephthalate)”. Please correct. Table 2 is illegible (specifically the 3rd column). Also Figure 1 and inscriptions under the Figure 3. Please correct. Page 14 line 5: “kg−1” – 1 should be in superscript. Please correct. For Figures and Tables, please use one style appropriate for this Journal (see page 16, lines 10, 11 and 23).
Response 1: We have modified the missing space between words or extra space. We abbreviate all "microplastics" of the manuscript to "MPs" and explain the abbreviations the first time we use them. We have modified the wrong symbol. We modified the wrong word. We used the IUPAC rules concerning polymer names to modify all polymer names in the manuscript and highlighted it in red in the revised manuscript. We selected the appropriate style for this journal to modify how the diagrams are marked in the manuscript.

Reviewer 3 Report
The paper studied the Microplastics from various point of view, which give us an overview about the Microplastics in terrestrial soils, so it should be published to make the other know about the accident related to this sport.
But the analysis and conclusion is hard to make out,for the record was not be well defined
Author Response
Response to Reviewer Comments
Thanks for your comments on our paper. We have revised our paper according to your comments. We hope that you find the revised manuscript acceptable for publication.
Point 1: The paper studied the Microplastics from various point of view, which give us an overview about the Microplastics in terrestrial soils, so it should be published to make the other know about the accident related to this sport.
But the analysis and conclusion is hard to make out,for the record was not be well defined.
Response 1: This manuscript chooses a newer method to identify microplastics in soil. At present, most instruments still have great challenges for identifying microplastics less than 10 μm, and there are few studies on the distribution and risk assessment of small microplastics in soils. However, the method used in this manuscript can quickly and accurately identify microplastics less than 10 μm. Therefore, this manuscript only records and analyzes the data of small microplastics with particle diameters between 0 and 35 μm in the study area, and performs environmental risk assessment of the small microplastics pollution in this area based on statistical data. At the same time, we hope that the statistical and risk assessment methods used in this manuscript can provide references for other researchers.
